# Snakebites in Cameroon by Species Whose Effects Are Poorly Described

**DOI:** 10.3390/tropicalmed9120300

**Published:** 2024-12-06

**Authors:** Jean-Philippe Chippaux, Yoann Madec, Pierre Amta, Rodrigue Ntone, Gaëlle Noël, Pedro Clauteaux, Yap Boum, Armand S. Nkwescheu, Fabien Taieb

**Affiliations:** 1MERIT Unit, Institut de Recherche pour le Développement, Paris Cité University, F-75006 Paris, France; 2Emerging Diseases Epidemiology Unit, Institut Pasteur, Paris Cité University, F-75015 Paris, France; 3Tokombere Hospital, Mora P.O. Box 74, Cameroon; amtapierre@yahoo.fr; 4Epicentre Yaounde, Yaounde P.O. Box 12069, Cameroon; rodriguentone@hotmail.com; 5Institut Pasteur, Translational Research Center, Paris Cité University, F-75015 Paris, France; gaelle.noel@gmail.com (G.N.); clauteauxpedro@gmail.com (P.C.); 6Institut Pasteur de Bangui, Bangui P.O. Box 923, Central African Republic; yap.boum2@pasteur-bangui.cf; 7Cameroon Society of Epidemiology, Yaounde P.O. Box 1411, Cameroon; nkwesch@yahoo.com; 8Institut Pasteur Medical Center, Paris Cité University, F-75015 Paris, France; fabien.taieb@pasteur.fr

**Keywords:** snakebite, envenomation, viper, elapid, antivenom, sub-Saharan Africa

## Abstract

Snakes responsible for bites are rarely identified, resulting in a loss of information about snakebites from venomous species whose venom effects are poorly understood. A prospective clinical study including patients bitten by a snake was conducted in Cameroon between 2019 and 2021 to evaluate the efficacy and tolerability of a marketed polyvalent antivenom. Clinical presentation during the first 3 days of hospitalization was recorded following a standardized protocol. This ancillary study aimed to assess the frequency of bites by the different species encountered in Cameroon and to describe the symptoms of bites by formally identified species. Of the 447 patients included in the study, 159 (35.6%) brought the snake that caused the bite that was identified by a specialist. Out of these, 8 specimens could not be identified due to poor condition, 19 were non-venomous species, and 95 belonged to *Echis romani*—formerly *E. ocellatus*—species. The remaining 37 specimens included 2 *Atheris squamigera*, 12 *Atractaspis* spp., 2 *Bitis arietans*, 11 *Causus maculatus*, 1 *Dendroaspis jamesoni*, 1 *Naja haje*, 1 *N. katiensis*, 5 *N. melanoleuca* complex, and 2 *N. nigricollis*. Symptoms, severity of envenomation, and post-treatment course are described. Symptoms and severity of bites are consistent with cases described in the literature, but some specific features are highlighted.

## 1. Introduction

Snakebite envenomation (SBE) is a major public health issue in sub-Saharan Africa (SSA). Recently added to the list of neglected tropical diseases (NTDs) by the World Health Organization (WHO), a strategy for the prevention and control of SBE has been defined to reduce mortality and disability by 2030 [1,2]. Each year, over 300,000 SBEs are treated in health facilities across sub-Saharan Africa (SSA), resulting in 10,000 deaths and as many permanent disabilities [3]. However, these figures are underestimated, and the reality is probably more than three times higher [3,4]. More than 95% of SBEs occur in rural areas, among the farming population, which largely explains the general lack of concern about snakebites despite their significant socioeconomic costs [5,6].

In Cameroon, three major families of venomous snakes are responsible for most accidents [7,8,9]. The Viperidae, mainly *Echis romani* (formerly *Echis ocellatus*), a potentially lethal species found in savannah, *Bitis*, several species of which are found throughout Cameroon, and *Atheris*, living in central and southern Cameroon, have an enzyme-rich venom that causes inflammation, bleeding disorders, and necrosis. *Causus maculatus*, on the other hand, is widespread and common but is not considered life-threatening to humans. The Elapidae, cobras of the genus *Naja,* present throughout Cameroon, and green mamba (*Dendroaspis jamesoni*) in the southern forest have a venom composed mainly of toxins causing postsynaptic paralysis, which leads to respiratory arrest, and of phospholipases A_2_ (PLA_2_), responsible for necrosis. As for the black mamba (*D. polylepis*), a particularly rare species in Cameroon, where only two specimens have been reported: one from Waza (Far North region) [10] and the other from near N’gaoundéré, Adamawa region [11]. The third group of snakes dangerous to humans is represented by the genus *Atractaspis* (nine species), which belongs to the family Lamprophiidae. These are burrowing snakes with mobile fangs in front of the jaw and a cytotoxic, hemorrhagic, and cardiotoxic venom. They are found throughout Cameroon. However, snakes responsible for bites are rarely identified, because few are brought to the hospital and health workers are not trained to identify them. As part of the clinical study “evaluation of antivenom in Africa” (Evaluation du Sérum Antivenimeux en Afrique, ESAA) conducted in Cameroon between 2019 and 2021 [12,13], we were able to identify the species of a large number of snakes responsible for bites and envenomations. The objectives of this study are to estimate the burden of bites by different species of venomous snakes in Cameroon and to describe the symptoms, severity, and outcome of bites by species whose SBE are poorly described, excluding those of *E. romani*.

## 2. Materials and Methods

The ESAA study was a prospective clinical survey in real conditions. Patients and methods are detailed elsewhere [12,13]. Briefly, patients were recruited between 25 October 2019 and 3 May 2021 in fourteen health centers across Cameroon (Figure 1). Inclusion criteria were (a) snakebite, with or without envenomation, (b) age equal to or greater than five years, (c) absence of known allergy to therapeutic serum of equine origin, (d) no administration of antivenom prior to hospital admission, and (e) signing informed consent. After inclusion, decisions to initiate antivenom and its dosage were made by the physician–investigator and based on the therapeutic algorithm recommended by the Cameroon Ministry of Health and the manufacturer’s guidelines. SBE grade assessment is detailed in Appendix A. A dry bite is a bite from a snake that has been identified as venomous but does not result in clinical signs. These victims do not receive antivenom.

We used Inoserp^TM^ PAN-AFRICA (IPA), manufactured by Inosan Biopharma (Mexico City, Mexico), currently the reference antivenom in Cameroon. It is a lyophilized polyvalent antivenom composed of highly purified fragments of immunoglobulins produced by immunizing horses with the venoms of fourteen species of snakes (*Echis ocellatus*, *E. pyramidum*, *E. leucogaster*, *Bitis gabonica*, *B. nasicornis*, *B. arietans*, *Naja haje*, *N. melanoleuca*, *N. nigricollis*, *N. pallida*, *Dendroapsis polylepis*, *D. viridis*, *D. angusticeps*, and *D. jamesoni*). According to the manufacturer, IPA has broad paraspecificity [14]. The antivenom dose is that recommended by the African Society of Venomology and the Cameroon Ministry of Health (A2). The dose is calculated according to the amount of venom the snake is likely to inject and the neutralizing capacity of the IPA. The double dose administered in the event of neurotoxic symptoms is explained by the rapidity of action of neurotoxins compared to the cytotoxic and hemotoxic components of venoms [12]. The antivenom administration protocol is detailed in Appendix B.

IPA was provided free of charge to all patients enrolled in the study; it was administered intravenously: 2 vials in cases of cytotoxicity/hemorrhage or 4 vials for neurotoxicity and renewed every two hours until improvement. It was administered either by slow direct intravenous route, lasting more than three minutes per 10 mL vial, or by infusion (two 10 mL vials of solution reconstituted and diluted in 50 mL of sterile isotonic saline) over thirty minutes, depending on the severity of symptoms and health personnel practices.

Data were first collected on case report forms (CRFs) and then entered into the REDCap data collection software versions 4.2.1 through 13.7.1 from study preparation to results analysis (Vanderbilt University, Nashville, TN, USA) [15,16]. Socio-demographic data, snakebite circumstances, time, and clinical presentation at hospital admission were collected. Patients were clinically assessed 2 h after the initial injection, and if new injections were needed, they were assessed again 2 h after each injection. IPA administration was repeated at the same dose in all patients with onset, persistence, or worsening of bleeding or neurotoxic disorders. Systematic evaluations were also scheduled 12, 24, 48 h, and 3 days after the first injection and at hospital discharge.

Patients were visited at home after hospital discharge to assess clinical progression and observe for signs of adverse reactions, including serum sickness.

The snake responsible for the bite was formally identified when the snake was brought to the hospital by the patient or family, or when a photo had been taken. Photos of the snake were sent to an expert (J.-P. Chippaux) to identify the species using published keys and descriptions [7,8].

An ethical clearance and Ministry of Health were obtained. In addition, formal written consent was obtained from all patients participating in the study, including children for whom consent was obtained from the parent/guardian who brought the child to the hospital.

## 3. Results

One hundred and fifty-nine patients brought the snake that had bitten them, eight of whom could not be identified due to their poor condition. The snake responsible for the bite was identified in 151 of 447 (33.8%) enrolled patients, including 95 (62.9%) *Echis romani*, 2 *Atheris squamigera*, 12 *Atractaspis* spp., 2 *Bitis arietans*, 11 *Causus maculatus*, 1 *Dendroaspis jamesoni*, 1 *Naja haje*, 1 *N. katiensis*, 5 *N. melanoleuca* complex, 2 *N. nigricollis*, and 19 species not dangerous to humans (Table 1). The distribution of the different species involved in each of the 14 hospitals where the study took place is shown in Figure 2.

### 3.1. Atheris squamigera (Table 2)

The two patients bitten by *Atheris squamigera*, an arboreal viper from the Central African Forest Region, were both from southern Cameroon.
tropicalmed-09-00300-t002_Table 2Table 2Symptoms on admission in patients bitten by *Atheris squamigera* (F = female; M = male).#Gender (Age)Time to Hosp. (h)EdemaWBCTBleedingNeuroNecrosis# Vials AVOutcomeAS1M (10)49:30Grade 2Grade 2Grade 4NoNo12RecoveredAS2M (32)4Grade 1NormalNoNoNo2Recovered


The first envenomation (AS1), a 10-year-old boy, was much more severe. He was bitten on his right forearm while sleeping in his bed at 2 a.m. Edema developed shortly after the bite. Hematemesis, estimated at 500 mL, prompted the patient to seek medical attention. He arrived at the hospital 2 days after the bite. He presented with regional edema extending beyond the nearest joint and incoagulable whole blood coagulation test WBCT (Figure 3). He vomited approximately 300 mL of hematemesis on admission. The hemorrhage decreased approximately ten hours after the first administration of IPA and had resolved by 30 h after admission to the hospital. He did not have renal failure. The patient was discharged on day 7 with discrete local edema not involving nearby joints. The child was seen 2 months later with no sequelae.

The other one (AS2), a 32-year-old man, was bitten on the hand at 6 a.m. in his home in Biyem Assi, a densely populated district of the city of Yaoundé (capital of Cameroon). The mild local edema decreased approximately 12 h after administration of 2 vials of IPA. He was discharged the same day. Two months later, he showed no sequelae.

### 3.2. Atractaspis spp. (Table 3)

Twelve patients were bitten by *Atractaspis*. In two, it was a dry bite (no envenomation). The remaining ten patients presented with mild to moderate cytotoxic envenomation (grade 1 or 2) without necrosis.
tropicalmed-09-00300-t003_Table 3Table 3Symptoms on admission in patients bitten by *Atractaspis* spp.: AT1 = *A. watsoni*; AT2 = *A. micropholis*; AT4 = *A. watsoni*; AT7 = *A. aterrima*; AT12 = *A. corpulenta*; AT3, AT5, AT6, AT8–11= unidentified species (F = female; M = male).#Gender (Age)Time to Hosp. (h)EdemaWBCTBleedingNeuroNecrosis# Vials AVOutcomeAT1F (32)1:30Grade 1NormalNoNoNo2RecoveredAT2F (8)4:20Grade 1Grade 2NoNoNo2RecoveredAT3F (16)0:25Grade 2NormalNoNoNo2RecoveredAT4M (45)0:20Grade 1Grade 2Grade 2NoNo2RecoveredAT5F (30)7:40Grade 2NormalNoNoNo2RecoveredAT6F (22)17Grade 2NormalNoNoNo2RecoveredAT7M (50)0:50Grade 2NormalNoNoNo2RecoveredAT8F (9)1:10Grade 2NormalNoNoNo2RecoveredAT9F (9)2Grade 1NormalNoNoNo2RecoveredAT10F (35)17:15NoNormalNoNoNo0Dry biteAT11F (11)2:30Grade 2NormalNoNoNo2RecoveredAT12M (12)2:20NoNormalNoNoNo0Dry bite


Two patients were bitten by *A. watsoni*. In the first one (AT1), the edema resolved in 48 h. In the second one (AT4), edema was trivial. On admission, patient AT4 presented with hemorrhage and incoagulable WBCT. Both resolved within 2 h of antivenom administration.

The WBCT of the patient bitten by *A. micropholis* (AT2) normalized two hours after admission.

On admission after a bite to the foot, patient AT5’s edema reached the knee (Figure 4). It began to decrease approximately 4 h after admission and was confined to the foot from the twelfth hour until discharge (day 3). He was examined a month later and showed no swelling or scarring.

In the patient bitten by *A. aterrima* (AT7), the edema appeared progressively during the first day and began to regress 24 h later. He was discharged on day 3. It was completely resolved by day 15.

The bite of *A. corpulenta* (AT12) was a dry bite, although the patient complained of left thoracic pain. Electrocardiogram was not performed due to lack of equipment.

### 3.3. Bitis arietans (Table 4)

The two *B. arietans* envenomations showed moderate edema without necrosis. There was no bleeding or WBCT abnormality. The edema resolved within 48 h in patient BA1 and within 24 h in patient BA2. Both were examined 16 and 12 days after the bite, respectively, and showed no edema or scarring.
tropicalmed-09-00300-t004_Table 4Table 4Symptoms on admission in patients bitten by *Bitis arietans* (F = female; M = male).#Gender (Age)Time to Hosp. (h)EdemaWBCTBleedingNeuroNecrosis# Vials AVOutcomeBA1F (24)0:10Grade 1NormalNoNoNo2RecoveredBA2M (31)17:15Grade 2NormalNoNoNo2Recovered


### 3.4. Causus maculatus (Table 5)

Five patients presented at the hospital with dry bites. They were discharged symptom-free after 4 h of observation. When they were re-examined 12 to 30 days later, none had any symptoms.

The other *C. maculatus* bites caused edema (six cases), in one case extending to the entire limb, but all without necrosis. Most edema resolves within 24 to 48 h. One patient had a local hemorrhage with normal WBCT that stopped before H2.
tropicalmed-09-00300-t005_Table 5Table 5Symptoms on admission in patients bitten by *Causus maculatus* (F = female; M = male; MD = missing data).#Gender (Age)Time to Hosp. (h)EdemaWBCTBleedingNeuroNecrosis# Vials AVOutcomeCM1F (8)16Grade 1NormalNoNoNo2RecoveredCM2M (63)7:30Grade 1NormalNoNoNo0RecoveredCM3F (10)MDNoNormalNoNoNo0Dry biteCM4M (27)1:15Grade 2NormalGrade 1NoNo2RecoveredCM5M (10)2NoNormalNoNoNo0Dry biteCM6F (80)0:20Grade 1NormalNoNoNo2Recovered CM7M (6)3:30NoNormalNoNoNo0Dry biteCM8F (27)1Grade 1Grade 1NoNoNo2RecoveredCM9M (40)1:50Grade 2NormalNoNoNo2RecoveredCM10F (28)0:45Grade 1NormalNoNoNo2RecoveredCM11M (11)0:30NoNormalNoNoNo0Dry bite


WBCT of patient CM8 showed incomplete coagulation, which normalized at H2.

The edema of patient CM9 increased between admission (approx. 3 h after the bite) and 4 h later (about 7 h after the bite) (Figure 5).

Patient CM10 was bitten twice by the same snake on her right thigh while sleeping in bed (Figure 6). The day after the bite, the edema increased, but the patient was discharged the same day.

### 3.5. Dendroaspis jamesoni (Table 6)

The patient had no local signs on admission. He complained of dysphagia a few minutes after the bite, without ptosis or muscarinic syndrome. Dysphagia resolved less than 4 h after administration of four vials of IPA.
tropicalmed-09-00300-t006_Table 6Table 6Symptoms on admission in the patient bitten by *Dendroaspis jamesoni* (M = male).#Gender (Age)Time to Hosp. (h)EdemaWBCTBleedingNeuroNecrosis# Vials AVOutcomeDJ1M (14)1:40NoNormalNoGrade 3No4Recovered


### 3.6. Naja haje (Table 7)

The patient bitten by *Naja haje* presented with local edema and blister (Figure 7) that worsened 5 h after admission despite antivenom treatment. There were no neurotoxic symptoms at any time. In addition to antivenom, symptomatic treatment was initiated according to the available drugs and the hospital’s own therapeutic practices. Symptomatic treatment included local care, in particular, puncture of the blisters, analgesics (paracetamol), and anti-inflammatories, including dexamethasone. She died 30 h after the bite of unknown cause, with no symptoms other than cytotoxic syndrome. It is possible that the delay in treatment (more than 16 h after the bite) reduced the efficacy of the antivenom and aggravated the cytotoxic syndrome.
tropicalmed-09-00300-t007_Table 7Table 7Symptoms on admission in the patient bitten by *Naja haje* (F = female).#Gender (Age)Time to Hosp. (h)EdemaWBCTBleedingNeuroNecrosis# Vials AVOutcomeNH1F (8)16:45Grade 3NormalNoNoNo2Died H12


### 3.7. Naja katiensis (Table 8)

Local edema resolved 4 h after IPA injection. However, it was replaced by local ecchymosis and extensive skin necrosis developed over the entire dorsum of the foot (Figure 8). The WBCT was normal and there was no bleeding or neurological deficit.
tropicalmed-09-00300-t008_Table 8Table 8Symptoms on admission in the patient bitten by *Naja katiensis* (F = female; M = male).#Gender (Age)Time to Hosp. (h)EdemaWBCTBleedingNeuroNecrosis# Vials AVOutcomeNK1F (60)5:30Grade 1NormalNoGrade 1Yes4Local scar


### 3.8. Naja melanoleuca (Table 9)

*N. melanoleuca* bites caused local edema (four out of five cases), which increased during the first 6 h after the bite and disappeared within 12 to 24 h after antivenom injection, except in the case of patient NM1, in whom the edema disappeared in about ten days (Figure 9). Two patients developed facial muscle paralysis (ptosis and mouth rictus), which improved 2 h (patient NM3) and 4 h (NM5; Figure 10), respectively, after administration of four vials of IPA. One patient (NM1) was 7 months pregnant and there were no fetal lesions.
tropicalmed-09-00300-t009_Table 9Table 9Symptoms on admission in patients bitten by *Naja melanoleuca*: NM1 = *N. subfulva*; NM2 to NM5 = *N. melanoleuca* (F = female; M = male).#Gender (Age)Time to Hosp. (h)EdemaWBCTBleedingNeuroNecrosis# Vials AVOutcomeNM1F (20)2:20Grade 2NormalNoNoNo2RecoveredNM2M (36)0:10Grade 1NormalNoNoNo2RecoveredNM3F (11)8:15Grade 2NormalNoGrade 3No4RecoveredNM4F (14)1:15Grade 1NormalNoNoNo2RecoveredNM5M (16)8:30NoNormalNoGrade 3No4Recovered


### 3.9. Naja nigricollis (Table 10)

Patient NN1 had a history of heart failure. The envenomation was moderate (grade 2 edema and no bleeding, no neurological signs at any time) and the immediate clinical course was favorable, with regression of symptoms allowing discharge on D2. Death of unknown cause on D20 was noted at the follow-up visit to assess late tolerance and evolution of the cytotoxic syndrome.
tropicalmed-09-00300-t010_Table 10Table 10Symptoms on admission in patients bitten by *Naja nigricollis* (F = female; M = male).#Gender (Age)Time to Hosp. (h)EdemaWBCTBleedingNeuroNecrosis# Vials AVOutcomeNN1F (80)4:10Grade 2NormalNoNoNo2Died D20NN2M (37)2:30NoNormalNoNoNo0Dry bite


## 4. Discussion

This ancillary study to the ESAA project made it possible to estimate the frequency of bites by the different snake species present in Cameroon (Table 1). It also enabled us to describe the symptoms and severity of SBE by species whose venom effects on humans are little known. The main limitation of this study is the small number of cases in each of the different species. However, this study benefits from (a) unbiased recruitment and absence of case selection, usually based on severity or specific features, and (b) standardized, accurate, and reliable clinical description and monitoring.

Here, the estimate of the frequency of bites by species whose envenomation is poorly described is based on one third of the snakebites recorded during this study and for which the snake was brought by the victim or his family. It is therefore an approximation but gives an interesting indication of the risk of being bitten by these different species. It was not always easy to identify the snake. The snake responsible for the bite is often used as a remedy by the traditional healer. According to some patients, the snake is disemboweled to obtain the entrails, which are applied to the bite and/or eaten by the victim to neutralize the venom. The snake is then burned and/or buried (Figure 11). Despite our best efforts, eight snakes could not be identified due to their poor condition, although it is most likely they were all colubrids.

On the other hand, the clinical follow-up, which was the main objective of the ESAA study [12,13], was reliable, resulting in precise clinical descriptions that were summarized here and will be commented on in the light of the literature. However, as with most hospital-based studies, there is a recruitment bias. Some patients with mild SBE do not go to the hospital and therefore avoid this type of evaluation. Conversely, severe cases, especially those who die quickly, do not have time to reach the hospital and are also ignored by the records.

### 4.1. Atheris squamigera

Envenomation by *A. squamigera* is characterized by high cytotoxicity, severe bleeding, and sometimes renal failure. The outcome can be fatal. The venom mainly contains PLA_2_, disintegrins, serine proteases, and metalloproteases, which explains the local and systemic symptoms. It activates fibrin formation, hydrolyzes fibrinogen, and induces platelet aggregation, resulting in thrombocytopenia [17,18]. There is currently no antivenom capable of specifically neutralizing the venoms of *Atheris* spp. Most polyvalent antivenoms covering African Viperidae species have very low neutralizing capacity [19], suggesting that very large amounts of antivenom would be required to neutralize *A. squamigera* venom.

Observations of six patients have been published. Three presented with mild or moderate envenomation, which was rapidly controlled by symptomatic treatment without antivenom [17,20,21]. Another patient presented with moderate envenomation and received 28 vials of different antivenoms (8 vials of IPA, 10 vials of Antivipmyn^®^ Tri, and 10 vials of Anavip^®^, the latter 2 antivenoms designed for American pitvipers) during the first 9 h, which did not prevent the progression of symptoms. However, clinical improvement occurred approximately 15 h after the onset of envenomation [22]. A fifth patient presented with regional edema and hemorrhagic syndrome 2 h after the bite. Local necrosis occurred 6 h after the bite. The patient received 35 vials of Near Middle East polyvalent antivenin (Chiron Behring Gmbh and Co) within 6 h of admission. Clinical and hematological symptoms began to normalize 10 h after antivenom administration and resolved 24 h after admission [23]. The last patient died of a hemorrhagic syndrome despite intravenous administration of 40 mL of Institut Pasteur polyvalent antivenom [24]. All these cases confirm the very poor neutralization of *A. squamigera* venom by currently marketed antivenoms. In any case, the dosages proposed by Ontiveros et al. [22] and Robinson et al. [23] are not feasible in SSA, where the patient must pay for all tests and treatments in the hospital, which would amount to more than a year’s family income.

### 4.2. Atractaspis sp.

Eight species of the genus *Atractaspis* are present in Cameroon (*A. aterrima*, *A. boulengeri*, *A. congica*, *A. corpulenta*, *A. dahomeyensis*, *A. irregularis*, *A. micropholis*, *A. reticulata*, *A.* wa*tsoni*). It was possible to identify the species only in 5 of the 12 treated patients (Table 3): *A. watsoni* (AT1 and AT4), *A. micropholis* (AT2), *A. aterrima* (AT7), *A. corpulenta leucura* (AT12). There are very few clinical descriptions in the literature, especially regarding these species. A comprehensive review of *Atractaspis* has recently been published [25]. However, with the exception of four cases of envenomation by *A. corpulenta* cited from the literature [26,27,28], no clinical data were available for the other three species.

Little is known about the composition of the venom, except for the sarafotoxins, which are structural and functional analogues of mammalian endothelins. They have a vasoconstrictive effect, particularly on coronary arteries. The venom also contains PLA_2_, metalloproteinases, Kunitz-type protein inhibitors, serine proteases, some of which degrade fibrinogen, proteins derived from β-defensins, cysteine-rich secretory proteins, and a wide variety of 3-finger toxins that are also abundant in elapid venoms [29,30,31,32]. The richness of *Atractaspis* venoms is consistent with the symptoms observed during envenomations. The venom composition explains, at least in part, the cytotoxic effects and bleeding.

*Atractaspis* bites are unique due to the anatomy of the skull of species belonging to this genus. The fangs are long and supported by a mobile, solenoglyph-like upper jaw, like that of the Viperidae. However, unlike the latter, the mouth cannot open sufficiently for the fangs to be erect and able to bite the prey or victim. The flexibility of the neck, combined with the habit of striking the closed mouth with a single fang, allows the latter to penetrate the flesh of the prey. The fang slides out through the corner of the mouth and the snake pulls its head back to ensure fang penetration. *Atractaspis* can wrap themselves around the prey to facilitate the insertion of the fang [33,34]. This feature suggests an adaptation to capture prey in confined spaces, such as burrows, or to bite into bedding. It is therefore more of a sting or scratch than a bite. As a result, the venom is generally inoculated in small amounts. Incidentally, this behavior and the way the victim is bitten often results in multiple bites—not to mention bites inflicted while handling the snake—which can sometimes explain the inoculation of larger amounts of venom [29,33,35,36]. On the other hand, some species have glands that are elongated towards the thorax and measuring up to 15 or 20% or even 33% of the body length, which can explain the injection of larger quantities of venom [25,26,37]. However, Cameroonian species appear to have short venom glands.

In most cases, envenomations by *Atractaspis* spp. are painful but moderate, except for a few cases of local necrosis. Some early symptoms (shortness of breath, cyanosis, precordial pain, abdominal cramps, vomiting, bradycardia or cardiac arrhythmia, hypertension) should attract attention, as they may reflect activation of endothelin receptors by sarafotoxins. It is sometimes difficult to distinguish these disorders from stress [28]. An electrocardiogram can confirm cardiac lesions, but this is not always possible in peripheral health centers, as in the case of patient AT12, who complained of precordial pain. It manifests as atrioventricular block on the electrocardiogram with bradycardia, arrhythmia, elevation of the S–T segment, prolongation of the P–R interval by more than 200 ms, and, in the most severe cases, intermittent disappearance of the QRS complex up to cardiac arrest [38,39]. Treatment consists of atropine administration or transcutaneous cardiac pacing if drug therapy fails.

A specific antivenom is manufactured by the National Antivenom and Vaccine Production Center in Riyadh, Saudi Arabia but is not available in sub-Saharan Africa. Bosentan, an inhibitor of endothelin receptors that are activated by sarafotoxins, has been shown to be more effective than the specific antivenom in protecting rabbits from a high dose of venom [40]. However, to our knowledge, bosentan has never been tested in humans.

In the literature, we found descriptions of 40 bites inflicted by one of the *Atractaspis* species present in Cameroon [25,27,33,34,35,36,37,41,42,43,44,45,46,47,48,49,50]. Among these, four died, including three from *A. irregularis*, twu male adults [33,35] and one infant [44], and one from *A. corpulenta* [28].

Of the 12 patients we treated, 8 were bitten at night, including 7 at home (i.e., inside the house), most of them while sleeping. This probably explains why 9/12 were females and 5/12 were children. Such proportions are not common in sub-Saharan Africa where most patients are young males aged 15–45 years [3]. We describe here for the first time SBEs—benign or moderate—by *A. micropholis* and *A. watsoni*.

### 4.3. Bitis arietans

The venom of this species consists of 79 proteins, including 23% serine proteases, 21% metalloproteinases, 11% C-type lectin, and 11% PLA_2_ [51].

Envenomation by *B. arietans* is stated to be severe. Visser and Chapman [52] reported a series of 210 *B. arietans* bites in Natal over a 7-year period, with 75 (35.7%) serious envenomations and 11 (5.2%) deaths despite treatment, some due to long delays in seeking medical attention. Seventeen cases of envenomation by *B. arietans* have been reported in the literature over the past 50 years. Significant cytotoxicity, often aggravated by extensive necrosis or gangrene, cardiovascular shock, and, more rarely, coagulopathy are the main symptoms described [52,53,54,55,56,57].

The two cases we treated were mild and responded well to antivenom administration, although one of them (BA2) had a delayed consultation

### 4.4. Causus maculatus

This small batrachophagous viper is synanthropic [58]. However, the venom is not very toxic to mammals in general and to humans in particular. Clinical descriptions are few and emphasize the importance of the inflammatory syndrome (pain, extensive edema, sometimes neutrophilic leukocytosis) but without necrosis, coagulopathy—except for rare bleeding from fang marks—or neurotoxicity [36,53,54,59,60].

### 4.5. Dendroaspis jamesoni

*D. jamesoni* venom contains a majority of 3-finger toxins (about 60% of total venom), including curare-like postsynaptic neurotoxins responsible for skeletal muscle paralysis and cytotoxins that depolarize cytoplasmic membranes nonspecifically, and Kunitz-type protein inhibitors (up to 20%). The remainder of the venom consists of PLA_2_, metalloproteinases, serine proteases, cysteine-rich secretory proteins, hyaluronidase, and natriuretic peptides [61].

We found only one article describing a lethal envenomation by *D. jamesoni* [59]. The venom is essentially neurotoxic, causing a postsynaptic block responsible for respiratory paralysis.

### 4.6. Naja haje

Over 50% of *N. haje* venom consists of 3-finger toxins, including neurotoxins and cytotoxins, and 25% of PLA_2_. Metalloproteinases, cysteine-rich secretory proteins, L-amino acid oxidases, Kunitz-type inhibitors, cobra venom factor, venom nerve growth factor, and other peptides have also been isolated from this venom [62,63].

There are very few descriptions of the envenomation of *N. haje*, the snake associated with Cleopatra’s suicide. The neurotoxicity of the venom with paralysis of the cranial nerves is responsible for the death of the patient by paralysis of the respiratory muscles [64,65]. However, one of the cases described by Warrell et al. [64] showed blisters, as in our patient (Figure 7). Zouari and Choyakh [65] successfully used intravenous neostigmine in their two patients to antagonize the venom neurotoxins and reverse the neuromuscular block.

The death of patient NH1 was surprising. He showed no signs of intolerance to IPA or neurological disorders. There was only cytotoxic envenomation. This explained the dose of two vials of IPA administered to the patient (instead of four vials in the case of neurotoxic envenomation), who was not considered to have been bitten by Elapid until the snake could be identified. At that time, the patient’s stable condition did not require antivenom administration. The Scientific Committee of the ESAA project investigated the possible causes of the death of patient NH1. It concluded that there was no evidence of intolerance after the first administration of IPA. Furthermore, the onset of severe local and systemic clinical symptoms 34 h after admission, when the patient appeared to be improving, suggests sepsis, inhalation of vomit, intoxication by conventional treatment, or pulmonary embolism. SBE evolution—and complication—or antivenom intolerance seems unlikely, although none of these causes can be completely ruled out [12].

### 4.7. Naja katiensis

*N. katiensis* venom consists of 50% 3-finger toxins, both neurotoxins and cytotoxins, and 25% PLA_2_. The remainder of the venom contains metalloproteinases, cysteine-rich secretory proteins, L-amino acid oxidases, Kunitz-type inhibitors, and some other peptides [62,66].

*N. katiensis* is a small spitting cobra, less than 1 m long, that projects its venom at a distance, especially into the eyes, causing very painful conjunctivitis without systemic envenomation [67].

To our knowledge, envenomation by *N. katiensis* has never been described. The symptoms are similar to those observed after a bite by *N. nigricollis* or *N. mossambica* [68,69].

### 4.8. Naja melanoleuca

The species *N. melanoleuca* has recently been revised [70]. Three species occur in Cameroon: *N. melanoleuca* throughout the southern forest region, *N. subfulva* in the savannah from the Adamaoua region to the Far North region, and *N. savannula* in the Far North region. Patient NM1 was bitten by *N. subfulva*, all others by *N. melanoleuca*. The recent separation of the species of the ‘*melanoleuca*’ complex could explain differences or even contradictions in the results regarding venom composition and envenomation symptomatology.

The majority of proteins isolated from *N. melanoleuca* venom are 3-finger toxins, including neurotoxins and cytotoxins (almost 60%) and PLA_2_ (about 15%). Other proteins include metalloproteinases (10%), cysteine-rich secretory proteins (almost 10%), and Kunitz-type inhibitors (less than 5%) [71,72]. Immunologically, the venom of species in the ‘*melanoleuca*’ complex is clearly distinct from that of other African najas, which has implications for antivenom production [73].

The three patients who received only two vials of antivenom showed no neurotoxic signs.

Very few clinical cases have been reported in the literature. Apart from the dry bite reported by Trape et al. [74], the other two published cases died of respiratory muscle paralysis due to the post-synaptic neurotoxins of the venom [36,75].

### 4.9. Naja nigricollis

*N. nigricollis* venom contains 40% 3-finger toxins and the same amount of PLA_2_. The rest of the venom is composed of numerous proteins, including metalloproteinases, cysteine-rich secretory proteins, L-amino acid oxidases, Kunitz-type inhibitors, etc. [63,66].

*N. nigricollis* is the large spitting cobra, more than 2 m long. As with *N. katiensis*, envenomation of the eyes results in highly inflammatory conjunctivitis [66]. The bite causes local inflammation often followed by extensive necrosis [68,76,77,78,79,80].

Our patient received only two vials of antivenom, as she showed cytotoxic and no neurotoxic signs.

Encounters with *N. nigricollis* are common, as this species is synanthropic. In a survey conducted in Nigeria in the 1970s, 106 people had been bitten by *N. nigricollis*—with 19% of the bites resulting in necrosis—and 40 had received the venom on any part of the body, including 26 in the eyes, with ocular lesions observed in 13 of these [77]. Warrell et al. [68] described 14 cases of envenomation by *N. nigricollis* in Nigeria, in an area close to that of our study. None showed neurotoxicity of the cranial nerves. However, all had edema involving the entire limb in more than half of the patients. Ten patients had quite extensive necrosis. In addition, three patients had coagulopathy, either due to platelet deficiency or fibrinolysis. Finally, 2 of 14 patients died because of envenomation.

The extensive tissue necrosis observed during envenomation by some *Naja* species is explained by the high levels of cytotoxins belonging to the 3-finger toxin family and PLA_2_ [66,81]. However, the concentration and activity of PLA_2_ in the venom are highly variable depending on the *Naja* species [82], which could explain the differences in the prevalence and extent of local lesions observed.

## 5. Conclusions

Although quite rare, SBEs from species whose venom effects are poorly understood should be considered with special care due to the challenges associated with treatment.

Even when antivenoms are available and effective, delayed hospital consultations can complicate outcomes. Polyvalent antivenoms, that are broadly paraspecific, are valuable but require larger doses. 

However, this may not always be feasible in peripheral hospitals lacking adequate resources and trained personnel. In such settings, symptomatic and adjuvant treatments remain critical, yet their provision poses significant challenges.

The large series of *Atractaspis* sp. bites suggests that exposure is relatively common, but envenomation is often less severe than expected. Similarly, our study confirms that *C. maculatus* bites are generally mild, with a high frequency of dry bite.

Envenomation caused by *D. jamesoni* and *Naja* spp. is rare. The symptomatology is more often inflammatory and cytotoxic than neurotoxic, depending on the type and concentration of the 3-finger venom. This inflammatory symptomatology is common in *N. nigricollis* and expected in *N. katiensis* but more surprising after *N. haje* bite. However, it is possible that some neurotoxic SBEs may have died before reaching the hospital.

These results underscore the need for enhanced training of community and healthcare workers in the detection and management of SBEs, particularly in affected regions with limited access to specialized care. To this end, the findings from this study can serve as valuable content for the development of digital tools, including tablet-based training modules, to equip healthcare and community workers with the necessary knowledge and skills to manage SBEs effectively. Such initiatives could significantly improve early management and overall outcomes in areas where SBE is a public health concern.

## Figures and Tables

**Figure 1 tropicalmed-09-00300-f001:**
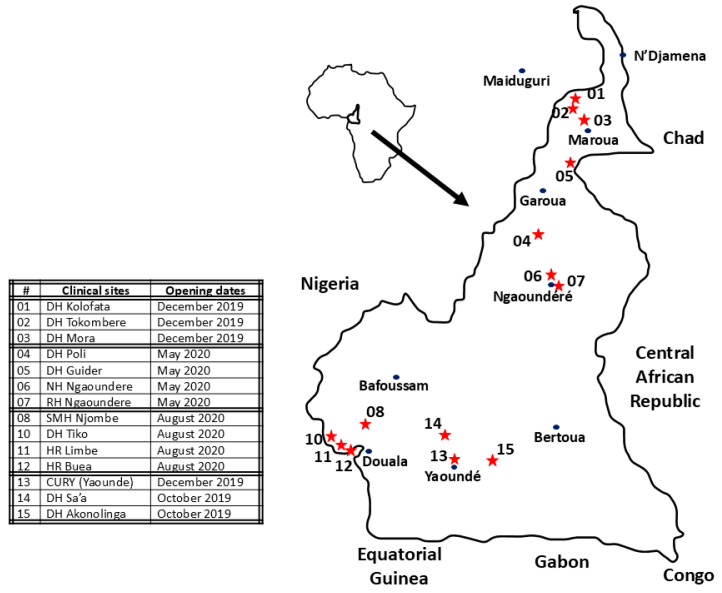
Location of centers participating in the ESAA study. The double inner borders define the geographical regions (01–03: Far North (Sahel); 04–07: North (Savannah); 08–12: West (Piedmont Forest) and 13–15: South (forest)). (Map redrawn from Wikipedia. https://fr.wikipedia.org/wiki/Cameroun#/media/Fichier:Cameroon_sat.png, CC0 1.0, accessed on 5 December 2024).

**Figure 2 tropicalmed-09-00300-f002:**
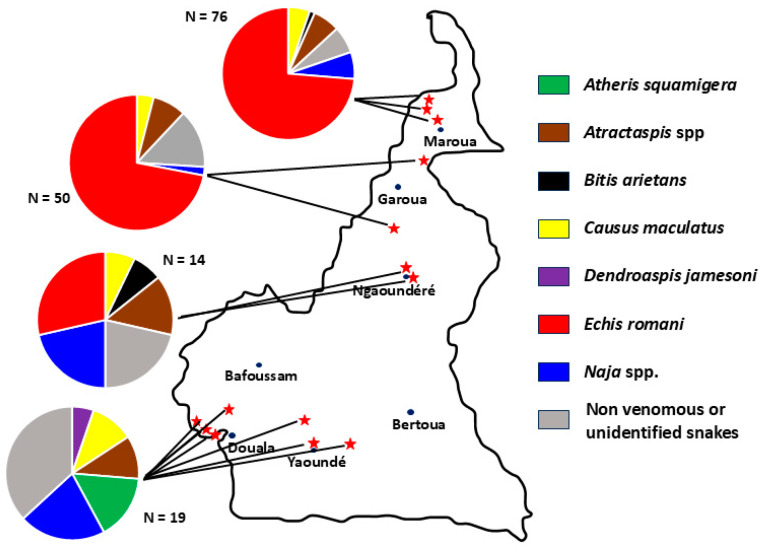
Identification of involved snakes according to the study sites (Map redrawn from Wikipedia. https://fr.wikipedia.org/wiki/Cameroun#/media/Fichier:Cameroon_sat.png, CC0 1.0, accessed on 5 December 2024).

**Figure 3 tropicalmed-09-00300-f003:**
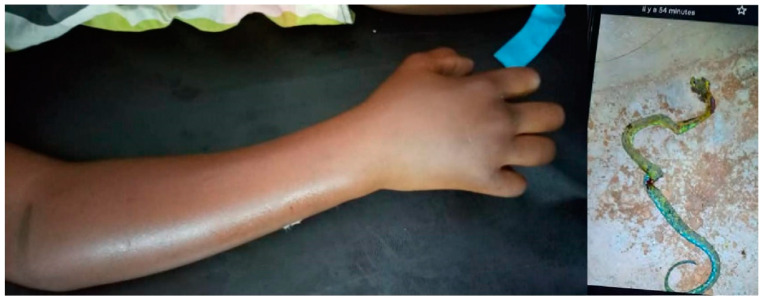
*Atheris squamigera* bite (AS2): swelling of the right forearm and photo of the snake sent from the family by telephone.

**Figure 4 tropicalmed-09-00300-f004:**
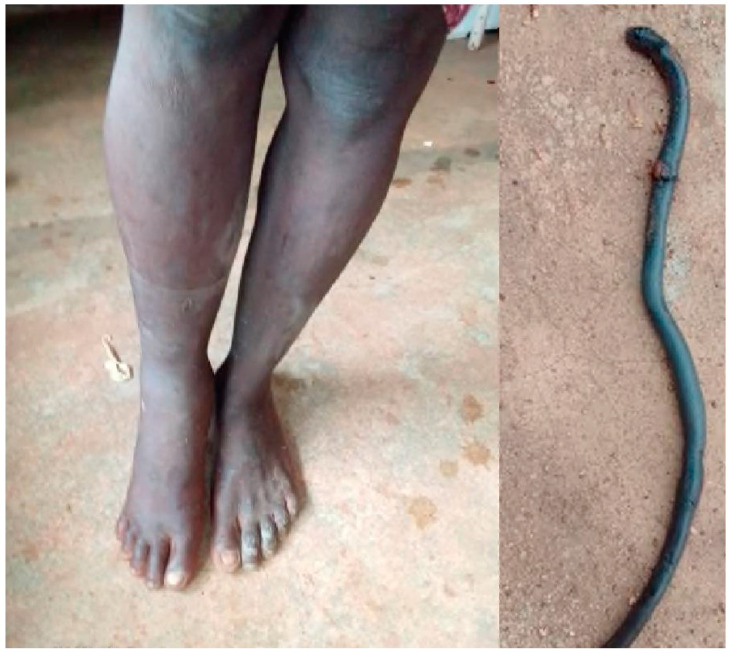
*Atractaspis* sp. bite (AT5): swelling of the right leg and photo of the snake brought by the family.

**Figure 5 tropicalmed-09-00300-f005:**
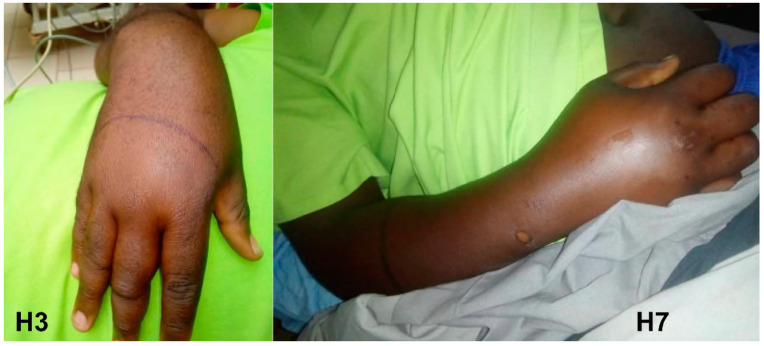
*Causus maculatus* bite (CM9): edema of the right forearm 3 and 7 h after bite, respectively, showing progression.

**Figure 6 tropicalmed-09-00300-f006:**
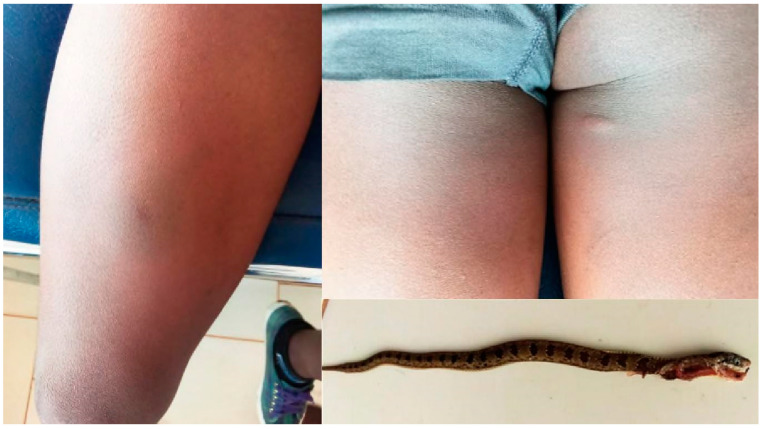
*Causus maculatus* bites in the same patient (CM10): edema of right thigh with circular hematoma at the first bite site on anterior thigh and small nodule at second bite site on posterior upper thigh; photo of snake brought by family.

**Figure 7 tropicalmed-09-00300-f007:**
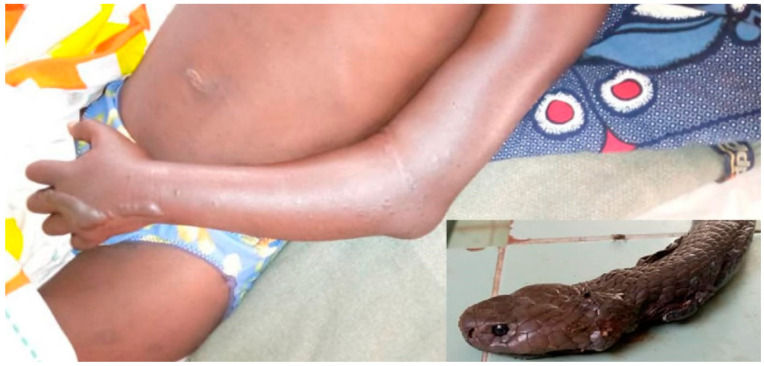
*Naja haje* bite (NH1): Edema of the left arm with blisters on the hand near the bite site and photo of the snake brought by the family.

**Figure 8 tropicalmed-09-00300-f008:**
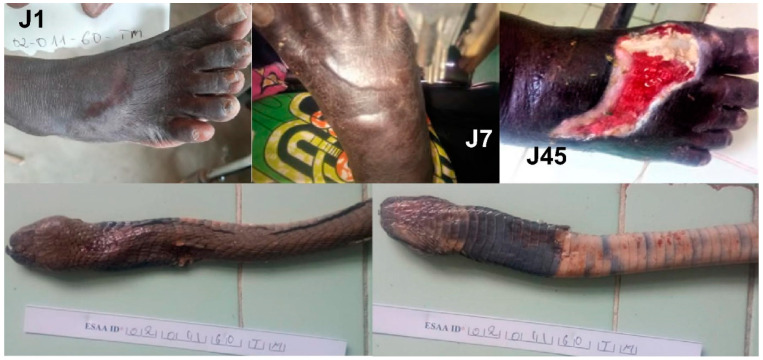
*Naja katiensis* bite (NK1): development of a necrosis on the dorsum of the right foot and photo of the snake brought in by the family.

**Figure 9 tropicalmed-09-00300-f009:**
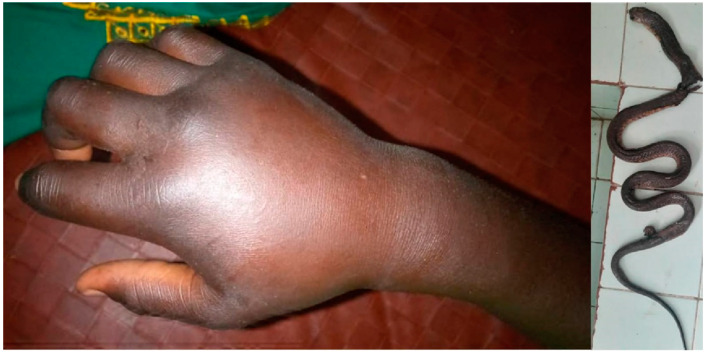
*Naja melanoleuca* (= *subfulva)* bite (Nm1): edema of the right hand at admission and photo of the snake brought in by the family.

**Figure 10 tropicalmed-09-00300-f010:**
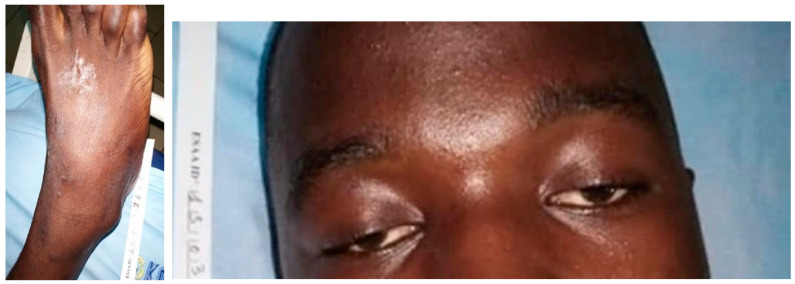
*Naja melanoleuca* bite (Nm5): poor local symptoms of the bite and ptosis.

**Figure 11 tropicalmed-09-00300-f011:**
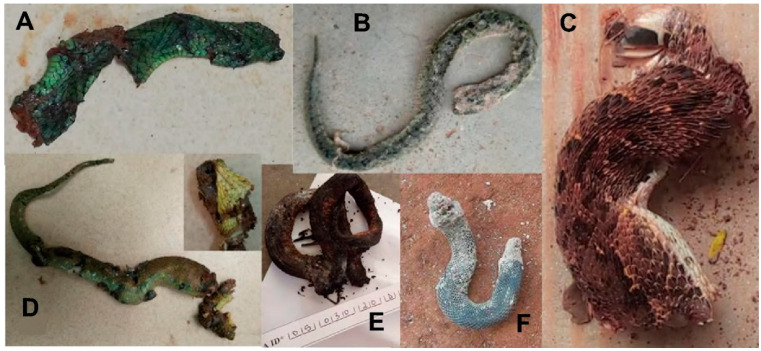
Snake remains brought in by the family after use by the traditional healer (see text). (**A**) = *Dendroaspis viridis*; (**B**) = *Bitis arietans*; (**C**) = *Echis romani*; (**D**) = *Atheris squamigera*; (**E**,**F**) = unidentified species.

**Table 1 tropicalmed-09-00300-t001:** Identification based on photos of the snake (N = 159).

Snake Species	Number, N (%)
**Boidae**	
* Eryx colubrinus*	2 (1.3)
**Colubridae**	
* Crotaphopeltis hotamboeia*	4 (2.5)
* Telescopus variegatus*	1 (0.6)
**Elapidae**	
* Dendroaspis jamesoni* *	1 (0.6)
* Naja haje* *	1 (0.6)
* Naja katiensis* *	1 (0.6)
* Naja melanoleuca* species complex *	5 (3.1)
* Naja nigricollis* *	2 (1.3)
**Lamprophiidae**	
* Atractaspis* spp. *	12 (7.5)
* Boaedon* spp.	5 (3.1)
* Psammophis* spp.	6 (3.8)
**Pythonidae**	
* Python sebae*	1 (0.6)
**Viperidae**	
* Atheris squamigera* *	2 (1.3)
* Bitis arietans* *	2 (1.3)
* Causus maculatus* *	11 (6.9)
* Echis romani* *	95 (59.7)
Unidentified species	8 (5)

* Dangerous venomous snakes for human.

## Data Availability

The clinical study data have been deposited in a publicly accessible repository and are available at DOI: 10.5281/zenodo.8200198 and DOI: 10.5281/zenodo.10609046, respectively. In the manuscript submitted to TropicalMed, some clinical details complement the general data available on the publicly accessible repository. They are extracted from the CRFs, which are not intended for online publication. If necessary, readers can ask us about aspects of interest to them.

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
