# Peer review of "Snakebites in Cameroon by Species Whose Effects Are Poorly Described"

_tropicalmed, 2024, doi:10.3390/tropicalmed9120300_

Round 1

Reviewer 1 Report

Comments and Suggestions for Authors

Dear Authors,

After carefully review the manuscript titled “Snakebites in Cameroon by species whose effects are poorly described”, the following points must be revised.

In general, please follow the guideline for preparing the figure and figure caption.  

Introduction:

1.     Line 61: please clarify the abbreviation ESAA

2.     Line 64/65 please delete a) and b)

Method: (in page 2)

1.     Severity of bleeding, necrosis, edema, WBCT and neurotoxicity would be indicated in the details for grading classification.

2.     There were 447 patients in this study but only 159 patients were included. How about the rest of 288 patients. Why were they excluded from the study? These exclusion criteria would be indicated.

3.     Criteria regarding IPA antivenom administration would be indicated in order to understand why some patients got antivenom administration but in some patients did not. Detail regarding standard amount of antivenom would let the reader understand the method and strategy of antivenom treatment of snake envenomation in Africa.

4.     Details regarding dry bite must be indicated.

Results:

1.     Table 1: the number of unidentified species (n = 8) must be included.

2.     Table 2 in page 5, data in the text and data in the table 2 are not consistent.

3.     Table 3, patients whom were bitten by unidentified species of Atractaspis spp. must be indicated.

4.     Line 257; N. melanoleuca must be italic (N. melanoleuca)

5.     Line 490; Naja nigricollis must be italic (Naja nigricollis)

6.     There were 2 fatal cases following envenomation caused by Naja spp. Was there any respiratory involvement?  Any other treatments for these 2 patients? Any involvement with the longer time to hospital?

7.     Line 235, please recheck she or he as data in table indicates the gender as female.

8.     Some Naja envenomed patients received 2 vials of antivenom which less than the indication as they did not present neurotoxicity. If the present of neurotoxicity appeared to be a crucial point to indicate the number of antivenom, this would be discussed.   

Author Response

Introduction:

  1. Line 61: please clarify the abbreviation ESAA

This refers to the study "evaluation of antivenom in Africa" (Evaluation du Sérum Antivenimeux en Afrique, ESAA).

- This sentence has been added to the text (lines 61-63).

  1. Line 64/65 please delete a) and b)

- This has been added to the text.

Method: (in page 2)

  1. Severity of bleeding, necrosis, edema, WBCT and neurotoxicity would be indicated in the details for grading classification.

- A sentence has been added referring to Appendix A which details the gradation of the different syndromes (line 79)

  1. There were 447 patients in this study but only 159 patients were included. How about the rest of 288 patients. Why were they excluded from the study? These exclusion criteria would be indicated.

- Of the 447 patients, only those who had brought the snake responsible for the bite were included. Of the 159 victims who brought the snake, it could only be identified in 151 patients. This is explained in the first paragraph of the results (lines 129-135).

  1. Criteria regarding IPA antivenom administration would be indicated in order to understand why some patients got antivenom administration but in some patients did not. Detail regarding standard amount of antivenom would let the reader understand the method and strategy of antivenom treatment of snake envenomation in Africa.

- A sentence has been added referring to Appendix A2 which details the antivenom administration protocol (lines 90-91

  1. Details regarding dry bite must be indicated.

- A dry bite is a bite from a snake that has been identified as venomous but does not result in clinical signs. These victims do not receive antivenom.

- This sentence has been added to the text (lines 79-81).

Results:

  1. Table 1: the number of unidentified species (n = 8) must be included.

- This has been taken into account and the percentages have been recalculated.

  1. Table 2 in page 5, data in the text and data in the table 2 are not consistent.

- Absolutely right. Thank you. We are sorry. The error has been corrected (lines 150-159)

  1. Table 3, patients whom were bitten by unidentified species of Atractaspis spp. must be indicated.

- This has been done (table caption).

  1. Line 257; N. melanoleuca must be italic (N. melanoleuca)

- This has been done.

  1. Line 490; Naja nigricollis must be italic (Naja nigricollis)

- This has been done.

  1. There were 2 fatal cases following envenomation caused by Najaspp. Was there any respiratory involvement?  Any other treatments for these 2 patients? Any involvement with the longer time to hospital?

Respiratory involvement corresponds to neurological syndrome grade 3 and above. In addition to the data in the text and tables, the following two sentences have been added for deaths due to Naja haje and N. nigricollis, respectively:

  • In addition to antivenom, symptomatic treatment was initiated according to the available drugs and the hospital's own therapeutic practices. Symptomatic treatment included local care, in particular puncture of the phlyctenes, analgesics (paracetamol) and anti-inflammatories, including dexamethasone. It is possible that the delay in treatment (more than 16 hours after the bite) reduced the efficacy of the antivenom and aggravated the cytotoxic syndrome (lines 258-264).

- The envenomation was moderate (grade 2 edema and no bleeding, no neurological signs at any time) and the immediate clinical course was favorable, with regression of symptoms allowing discharge on D2. Death of unknown cause on D20 was noted at the follow-up visit to assess late tolerance and evolution of the cytotoxic syndrome (lines 303-307).

  1. Line 235, please recheck she or he as data in table indicates the gender as female.

- Absolutely right. Thank you. The error has been corrected (line 261)

  1. Some Najaenvenomed patients received 2 vials of antivenom which less than the indication as they did not present neurotoxicity. If the present of neurotoxicity appeared to be a crucial point to indicate the number of antivenom, this would be discussed.

The patients bitten by elapids who received 2 ampoules instead of 4 were all patients without respiratory disorders but with cytotoxic syndrome. According to the protocol (Appendix A2), they received 2 vials of antivenom. This point is mentioned in the discussion.

- For the patient bitten by Naja haje, this is already explained lines 479-490.

- For the patients bitten by N. melanoleuca, a sentence was added : "The 3 patients who received only 2 vials of antivenom showed no neurotoxic signs" (lines 5182).

- For the patients bitten by N. nigricollis, a sentence was added : "Our patient received only 2 vials of antivenom, as she showed cytotoxic and no neurotoxic signs" (lines 530-531).

Reviewer 2 Report

Comments and Suggestions for Authors

This is a useful case series and very interesting. There is some tint of advertising the reference antivenom and I think this is most evident in lines 84-86. The reference itself is simply from a conference does not lead to the publication by Mathe but actually to a conference. This will need to be fixed. The paper in reference is a self-review of the antivenom by the manufacturer to support the claim.  Since the paper is not about antivenom or the antivenom product I suggest removing or heavily revising this sentence and that should address my ethical concern. It might be true, but it is not the focus of the paper and looks like a product claim. 

However, due to its broad paraspecificity, IPA neutralizes the 83 venom of a large number of SSA venomous snake species, including some other than 84 those used in its manufacture, with variable efficacy [14]. 

Author Response

This is a useful case series and very interesting. There is some tint of advertising the reference antivenom and I think this is most evident in lines 84-86. The reference itself is simply from a conference does not lead to the publication by Mathe but actually to a conference. This will need to be fixed. The paper in reference is a self-review of the antivenom by the manufacturer to support the claim.  Since the paper is not about antivenom or the antivenom product I suggest removing or heavily revising this sentence and that should address my ethical concern. It might be true, but it is not the focus of the paper and looks like a product claim. 

However, due to its broad paraspecificity, IPA neutralizes the 83 venom of a large number of SSA venomous snake species, including some other than 84 those used in its manufacture, with variable efficacy [14]. 

- We thank the reviewer for his/her positive opinion.

- We have substantially modified the sentence: "According to the manufacturer, IPA has broad paraspecificity [14]".

Reviewer 3 Report

Comments and Suggestions for Authors

This is a very interesting and well-written paper. The authors provide new information about the various snakes in Cameroon and their patients' response to the regional antivenoms. This is valuable information that will improve medical care in this region of Africa.

there are a few minor typoes:

Line 112  Health

Line 481   neurotoxins

Line 490. Italics for Naja nigricollis

Author Response

This is a very interesting and well-written paper. The authors provide new information about the various snakes in Cameroon and their patients' response to the regional antivenoms. This is valuable information that will improve medical care in this region of Africa.

- We thank the reviewer for his positive opinion.

there are a few minor typoes:

Line 112  Health

- This has been done (line 117).

Line 481   neurotoxins

- This has been done (line 506).

Line 490. Italics for Naja nigricollis

- This has been done (line 516).

Reviewer 4 Report

Comments and Suggestions for Authors

The manuscript provides valuable and detailed information about snakebite envenomation in Cameroon, focusing on species whose venom effects are not well-known. This is an important topic in tropical medicine and public health, especially in sub-Saharan Africa. The study helps fill gaps in knowledge by carefully documenting snakebite cases and their clinical effects.

The study highlights snake species that are often overlooked in the literature. The findings give us a better understanding of snakebite epidemiology and clinical symptoms in sub-Saharan Africa.

The multi-center design adds depth and diversity to the data. Including 159 cases is impressive, and the use of photos validated by experts makes the findings very reliable.

The post-discharge home visits to monitor patient recovery and look for adverse reactions, such as serum sickness, are a great addition. This approach provides valuable details and sets a good example for future studies.

The manuscript mentions the use of Inoserp™ PAN-AFRICA antivenom, but it would be helpful to explain the rationale for the dosing—why two vials for cytotoxic or hemorrhagic cases and four for neurotoxic cases? More details about whether this is based on clinical evidence or practical considerations would improve the discussion.

The study reports a rare fatality from Naja haje envenomation but does not provide enough detail on the cause. Could it have been due to neurotoxicity, infection, or complications like tissue necrosis? Further discussion on this case would add important insights.

It would also be helpful if the manuscript identified gaps in knowledge more clearly. For example, how much does the lack of species-specific antivenoms affect treatment outcomes?

The rationale for antivenom dosing should be explained further. Are the doses based on specific species, severity, or availability? A more detailed discussion would enhance the manuscript.

The fatality involving Naja haje should be discussed in more depth. Were sepsis, vomit inhalation, or other complications like pulmonary embolism considered? This additional analysis would make the case clearer.

Tissue necrosis is commonly reported in Naja spp. envenomation. Were there cases of necrosis observed in bites from Naja melanoleuca or Naja nigricollis? This aspect could be explored further.

This study makes a meaningful contribution to our understanding of snakebite envenomation in Cameroon, especially for lesser-studied species. The study is well-designed, and the methods are robust. However, including more discussion about antivenom use, causes of death, and public health implications would further strengthen the manuscript and make it even more impactful for clinicians, researchers, and public health professionals.

Author Response

The manuscript provides valuable and detailed information about snakebite envenomation in Cameroon, focusing on species whose venom effects are not well-known. This is an important topic in tropical medicine and public health, especially in sub-Saharan Africa. The study helps fill gaps in knowledge by carefully documenting snakebite cases and their clinical effects.

The study highlights snake species that are often overlooked in the literature. The findings give us a better understanding of snakebite epidemiology and clinical symptoms in sub-Saharan Africa.

The multi-center design adds depth and diversity to the data. Including 159 cases is impressive, and the use of photos validated by experts makes the findings very reliable.

The post-discharge home visits to monitor patient recovery and look for adverse reactions, such as serum sickness, are a great addition. This approach provides valuable details and sets a good example for future studies.

  • We would like to thank the reviewer for his very positive and kind review.

The manuscript mentions the use of Inoserp™ PAN-AFRICA antivenom, but it would be helpful to explain the rationale for the dosing—why two vials for cytotoxic or hemorrhagic cases and four for neurotoxic cases? More details about whether this is based on clinical evidence or practical considerations would improve the discussion.

- The treatment protocol is that of the African Society of Venimology, endorsed by the Ministry of Health of Cameroon (as well as several ministries of health in sub-Saharan countries). It was established after numerous studies since 2007, which would take too long to detail here. However, the protocol has been included as an appendix (A2) and the relevant references have been provided.

The study reports a rare fatality from Naja haje envenomation but does not provide enough detail on the cause. Could it have been due to neurotoxicity, infection, or complications like tissue necrosis? Further discussion on this case would add important insights.

  • Details have been added in the Results chapter (lines 258-261), in addition to what is explained in the Discussion chapter (lines 479-490).
  • "In addition to antivenom, symptomatic treatment was initiated according to the available drugs and the hospital's own therapeutic practices. Symptomatic treatment included local care, in particular puncture of the phlyctens, analgesics (paracetamol) and anti-inflammatories, including dexamethasone. […] It is possible that the delay in treatment (more than 16 hours after the bite) reduced the efficacy of the antivenom and aggravated the cytotoxic syndrome." (lines 258-264).

It would also be helpful if the manuscript identified gaps in knowledge more clearly. For example, how much does the lack of species-specific antivenoms affect treatment outcomes?

- The results of treatment would probably not be affected by the use of specific antivenoms. However, on the one hand, such antivenoms do not exist and it is therefore impossible to assess their potential impact, and on the other hand, their absence is due to the inability of health workers to identify snakes due to lack of appropriate training (see lines 60-61) and to the poor condition of snakes reported by victims (lines 323-329).

The rationale for antivenom dosing should be explained further. Are the doses based on specific species, severity, or availability? A more detailed discussion would enhance the manuscript.

- An explanatory sentence has been added to lines 90-95.

- The antivenom dose is that recommended by the African Society of Venomology and the Cameroon Ministry of Health (A2). The dose is calculated according to the amount of venom the snake is likely to inject and the neutralizing capacity of the IPA. The double dose administered in the event of neurotoxic symptoms is explained by the rapidity of action of neurotoxins compared to the cytotoxic and hemotoxic components of venoms [12].

The fatality involving Naja haje should be discussed in more depth. Were sepsis, vomit inhalation, or other complications like pulmonary embolism considered? This additional analysis would make the case clearer.

  • This was done on lines 479-490.

Tissue necrosis is commonly reported in Naja spp. envenomation. Were there cases of necrosis observed in bites from Naja melanoleuca or Naja nigricollis? This aspect could be explored further.

  • We have described what was observed with the different Naja species ( haje, lines 258-264; N. katiensis, lines 273-275; N. melanoleuca, lines 287-293; and N. nigricollis, lines 303-307) as well as in the discussion for some of these species (N. haje, lines 479-490; N. nigricollis, lines 541-545).

This study makes a meaningful contribution to our understanding of snakebite envenomation in Cameroon, especially for lesser-studied species. The study is well-designed, and the methods are robust. However, including more discussion about antivenom use, causes of death, and public health implications would further strengthen the manuscript and make it even more impactful for clinicians, researchers, and public health professionals.

  • The reviewers' criticisms have all been taken into account, and we thank them for their relevant comments.